# Musculoskeletal Fitness for Identifying Low Physical Function in Older Women

**DOI:** 10.3390/ijerph20085485

**Published:** 2023-04-12

**Authors:** Frederico Abreu, Vera Zymbal, Fátima Baptista

**Affiliations:** 1Department of Sports and Health, CIPER, Faculdade de Motricidade Humana, Universidade de Lisboa, 1495-751 Lisbon, Portugal; 2ESS, Instituto Politécnico de Setúbal, 2910-761 Setúbal, Portugal

**Keywords:** jump power, handgrip strength, physical activity, daily steps, ageing

## Abstract

Aims: This cross-sectional study aimed to analyze the relevance of musculoskeletal fitness for identifying low physical functioning in community-dwelling older women. Methods: Sixty-six older women (73.62 ± 8.23 yrs old) performed a musculoskeletal fitness assessment of the upper and lower limbs. A handheld dynamometer was used to evaluate upper-limb muscle strength through a handgrip (HG) test. Lower-limb power and force were assessed from a two-leg countermovement vertical jump (VJ) on a ground reaction force platform. Physical functioning was assessed subjectively using the Composite Physical Function (CPF) questionnaire and objectively by daily step count measured by accelerometry and gait speed/agility assessed by the 8-Foot Up-and-Go (TUG) test. Logistic regressions and ROC curves were carried out to define odds ratios and ideal cutoff values for discriminatory variables. Results: VJ power showed the ability to identify low physical functioning when evaluated through the CPF (14 W/kg, 1011 W), gait speed/agility (15 W/kg, 800 W), or daily accumulated steps (17 W/kg). Considering that VJ power was normalized for body mass, the increase of 1 W/kg corresponds to a decrease of 21%, 19%, or 16% in the chance of low physical functioning when expressed by these variables, respectively. HG strength and VJ force did not show a capacity to identify low physical functioning. Conclusions: The results suggest that VJ power is the only marker of low physical functioning when considering the three benchmarks: perception of physical ability, capacity for mobility, and daily mobility.

## 1. Introduction

Data from the World Health Organization suggest that population ageing is evolving more rapidly than in the past. From 2015 to 2050 the proportion of individuals aged 60 years and older will nearly double [1].

Ageing is associated with a long list of decaying physiological systems—cardiovascular, endocrine, neuromuscular, and others [2]—which can result in a loss of musculoskeletal capacity [3,4], particularly muscle strength and power that determines a greater or lesser extent the physical function [5]. Physical function is a person’s ability relative to the physical–social context where physical functioning occurs [6]. The loss in musculoskeletal capacity is of high importance later in life as it determines the ability to perform and function daily independently [7]. In addition, it is a central part of the mechanism through which frailty and sarcopenia can affect older people’s lifespan, health, and independence [5,6]. Autonomy and independence rely on the ability to perform basic and instrumental daily tasks, as physical function loss leads to the onset of dependence and disability [8]. Hence, the importance of evaluating musculoskeletal fitness as a proxy for physical functioning is increasing in research and clinical settings [7]. Here we define musculoskeletal fitness as the capacity of the musculoskeletal system to produce force, considering both the absolute maximal capacity (strength) and the force–time relationship (power). In this regard, some difficulties arise regarding which musculoskeletal fitness parameters should be assessed better to represent the level of physical functioning in older age. The evaluation of these parameters is of high relevance in many contexts. First, in clinical settings, a cascade of personal, social, and clinical consequences follow the loss of physical function. Both general practitioners and geriatricians would take advantage of specific and sensitive tools that would enhance the quality and efficacy of opportunistic screening in routine consultations [7]. Moreover, community programs could target the specific limiting factors of physical function in multicomponent interventions.

Handgrip strength has been massively used as a marker of global body strength [9] due to its practicality, low cost, and validity throughout all age groups, especially in old age [10]. Handgrip strength is essential in many activities of daily living so that one can feel safe when their gait is slightly compromised and handle necessary instruments and tools throughout the day. Although some research suggests that handgrip strength increases may be limited in adult age [11] and may not express the potential functional gains obtained from an exercise program in older adults [12], it is still the most common strength assessment in this population. More ecological approaches, such as strength assessment of compound movements (sit-to-stand, push, and pull), require more technological resources and may also present a greater risk of injury for those who still do not have limitations in performing these tasks. The fact that there is virtually no technical demand besides the familiarization with the dynamometer and the instruction to “squeeze” makes the handgrip strength test the go-to assessment in older individuals.

To date, many studies have reported that the muscle power of the lower limbs appears to be a stronger predictor of physical functioning in older adults than handgrip strength [13,14]. While the capacity to produce a significant amount of maximal force is essential, it may not be the most limiting factor regarding the performance of daily tasks. Compared to muscle strength, muscle power has shown a more remarkable ability to explain the variability of physical functioning resulting from ageing [14]. Getting up from a chair, crossing a road, climbing stairs, or reestablishing posture after an imbalance depends more on the ability to produce force quickly rather than how much force can be produced. Moreover, a recent study has suggested muscle power of the lower limbs as a potential mediator of the effects of physical activity on physical functioning [15]. It is known that regarding musculoskeletal quality, the ageing process induces a greater loss of the muscle fiber type specifically responsible for power output [3,4]. In this line, an international position statement refers to muscle power training as the most effective when the goal is to translate into functional improvements in daily living activities [8]. Currently, the muscle power of the lower limbs of older people is assessed using either jumping or standing up and sitting down from a chair/bench on force platforms or is estimated from predictive equations [16].

Given that muscle strength and power determine the performance of the activities of daily living in later life, this study aimed to investigate their relevance and ability to discriminate between older women with and without low physical functioning. Many efforts have been made to identify variables and values for suspecting or identifying geriatric syndromes in which musculoskeletal fitness is crucial, such as sarcopenia, frailty, or risk of falls, which are risky syndromes for physical functioning. In this study, we intended to identify variables/cutoff values for the suspected risk of low physical functioning per se. We assume different variables/cutoff values for musculoskeletal fitness, depending on the reference variables for physical functioning: self-reported or performance (ability and behavior).

## 2. Materials and Methods

### 2.1. Data Collection

The sixty-six female participants included in this study were selected from a convenience sample of 671 young, adult, and older people recruited to characterize the physical fitness and physical activity of residents in Lisbon, Portugal. Participants were voluntarily recruited from the Lisbon city area in response to leaflets and posters distributed around the community senior centers. Eligibility was conferred if the volunteer was 65 or older and resided in the community. Participants who could not to perform any tests due to severe limitations or other health issues were automatically excluded. The study was approved by the Faculty of Human Kinetics Ethics Committee, University of Lisbon, and conducted in agreement with the Helsinki Declarations. All participants provided informed consent.

### 2.2. Musculoskeletal Fitness

Musculoskeletal fitness was assessed in the upper and lower limbs. A handheld dynamometer was used to evaluate upper-limb muscle strength through handgrip maximum isometric contraction (Jamar, Lafayette, IN, USA). In the seated position with the unsupported dominant arm’s elbow at 90° flexion, participants were asked to perform three maximal handgrip contractions with a rest interval of 2 min. The trained staff requested that the participants maintain their body position while gripping as hard as possible. The highest obtained value was recorded. The absolute maximum recorded value was then divided by the participant’s body mass to obtain the relative handgrip strength.

Lower-limb maximum power and maximum force were assessed using a ground reaction force platform (Leonardo Mechanograph, Novotec Medical, Pforzheim, Germany). Participants were asked to perform a single two-legged countermovement jump (s2LJ) as high as possible using both legs, land on the forefoot, and stand as still as possible on both feet. For this purpose, a jumping position was assumed, with the hands resting at the waist and the feet at the width of the pelvis. Participants performed three jumps: one practice jump and two test jumps (~30 s apart). The highest jump was analyzed using Leonardo software version 4.4 (Novotec Medical, Pforzheim, Germany). Two trained staff members were present during the jumps to ensure safety and help participants regain balance if necessary.

### 2.3. Physical Functioning

Physical functioning was assessed using three approaches: a subjective approach to people’s perception of their capacity to perform activities of daily life; an objective approach to the capacity itself expressed by gait speed/agility; and a behavioral approach regarding habitual mobility (physical activity) expressed by the number of steps accumulated daily.

The participant´s perception of their capacity to perform activities of daily living was assessed through subjective self-report using the 12-item Composite Physical Function (CPF) scale [17,18]. This tool assesses the physical function across a variety of domains, such as basic ADLs (e.g., bathing and walking 1 or 2 blocks), instrumental ADLs (e.g., shopping and carrying some weight), and advanced activities (e.g., vigorous sports). Each of the 12 items is attributed a score ranging from 0 to 2 respective to each response (0 = cannot do, 1 = can do with some help, 2 = can do independently without help). The sum of the partial scores represents an overall score.

To assess gait speed/agility, participants were asked to perform the 8-Foot Up-and-Go test (TUG) as described by Jones and Rikli [17]. Sitting on a 43 cm armless chair with their hands on their knees and feet flat on the floor, individuals had to stand up and walk as fast as possible around a cone set 2.44 m away, then return to the chair and the sitting position. Two attempts were made, and the fastest time was recorded. Despite being a test to measure agility, its results depend primarily on the participant’s gait speed and may serve as a proxy for walking slowness.

Daily mobility represented by daily steps was assessed by accelerometry (Actigraph, model GT3X, Pensacola, FL, USA). Participants were requested to wear the device on their right hip for four consecutive days, divided equally by weekdays and weekend days. During sleep or water-based activities, the accelerometers were taken off. The devices collected the movement counts in raw mode—100 Hz frequency—and then the data were downloaded into 15 s epochs (Actilife v.6.9.1).

All approaches to assess musculoskeletal fitness and physical functioning have demonstrated excellent validity (ICC > 0.90) and/or reproducibility (r > 0.75) [19,20,21].

### 2.4. Statistical Analysis

Collected data were analyzed using SPSS statistical software package (version 28 for Windows; SPSS, Chicago, IL, USA). Descriptive statistics, such as means ± standard deviation (SD), were calculated for all variables. Pearson’s correlation analysis was performed to determine the association between normally distributed variables and Spearman’s correlation analysis for skewed variables. Logistic regressions were performed to analyze the odds ratios and determine risk reductions in low physical function, low walking speed/agility, and low daily mobility for every independent variable (jumping power, force, and handgrip strength). Low physical function was determined by a score of ≤14 points [17]; low gait speed/agility, by a cutoff time of 8 s as suggested [18]; and low mobility was determined by a daily step count average lower than 4600 [22].

ROC analyses were used to assess ideal cutoff values for vertical jump power, jump force, and handgrip strength as potential predictors of low physical function, low walking speed, and low physical activity. The area under the curve (AUC) was calculated to determine the cutoff value’s capability as a screening tool. Given the statistical significance (*p* < 0.05), the best trade-off between specificity (sp) and sensitivity (se) was found in order to maximize both.

## 3. Results

The characterization of the sample is presented in Table 1: anthropometric characteristics, physical functioning evaluated through CPF, gait speed/agility, daily steps, and the musculoskeletal fitness expressed through the handgrip strength, vertical jump power, and vertical jump force. Low physical function was observed in ~11% of participants when the criterion was CPF, ~14% of participants when the criterion was gait speed/agility, and 21% when the criterion was daily accumulated steps.

Correlation analysis showed associations of upper-body muscle strength with CPF and TUG but not with daily steps (Table 2). Lower-body muscle power was found to correlate with all three measures of physical functioning. In addition, physical functioning correlation coefficients were higher with lower-limb muscle power than with handgrip strength. Lower-limb muscle force was not related to any measures of physical functioning.

The cutoff values, percentiles (P), classification accuracy, and odds ratio used to identify low physical functioning by handgrip strength, vertical jump power, and vertical jump force are described in Table 3. Only models with vertical jump power with adjustment for body mass (relative power, W/kg) but also without adjustment (absolute power, W), showed reasonable discrimination (AUC: 0.73–0.79) and predictability (Se and Sp: 64.3–89.5%) for low physical functioning. Specificity was equal to or greater than sensitivity; that is, the models showed a remarkable ability to identify those who do not have low physical functioning (negative cases). 

Regarding relative vertical jump power, the cutoff values correspond to 13.88 W/Kg (P24; AUC 0.79; sig 0.013; Sp 81.4%; Se 71.4%) for the identification of low physical function, 15.18 W/Kg (P35; AUC 0.74; sig 0.023; Sp 70.2%; Se 66.7%) for the identification of low gait speed/agility, and 17.13 W/Kg (P44; AUC 0.73; sig 0.011; Sp 65.1%; Se 64.3%) for the identification of low daily mobility. Considering the absolute vertical jump power, the identification cutoff values were 1011W (P32; AUC 0.73; Sp 72.9%; Se 71.4%) for low physical function and 800W (P18; AUC 0.78; Sp 89.5%; Se 66.7%) for low gait speed/agility.

The logistic regression showed that a 1 W/kg increase in relative jumping strength in older women was associated with a 21% reduction in the chance of low physical functioning when evaluated with CPF (<14 pts), 19% reduction when evaluated through gait speed/agility (TUG > 8 s), and 16% reduction when evaluated by habitual physical activity (<4600 steps/day) (Table 3, Figure 1).

Considering absolute values, the reduction in the risk of low physical functioning was 3% when evaluated through the CPF and 5% when evaluated through the gait speed/agility for each 10 W increase in vertical jump power.

## 4. Discussion

The ability to maintain high functioning levels in old age, despite chronic disease, has been linked to preservation of skeletal muscle function [7,8,23]. Hence, the fact that most activities of daily living are dependent on muscle function means that maintaining physical independence and autonomy later in life relies significantly on one’s skeletal muscle capacity. This study aimed to analyze the relevance of musculoskeletal fitness to identify low physical functioning in community-dwelling older women. For this purpose, three measures of musculoskeletal fitness (handgrip, VJ power, and VJ force) and three measures of physical functioning were considered (CPF, TUG, and daily mobility). Associations were analyzed, regardless of the main goal being centered on identification analyses.

In agreement with most studies, our results suggest that handgrip strength and lower-limb muscle power are associated with physical functioning in ADLs [24,25,26]. Considering all measures’ capacity to identify physical functionality, only the VJ power could differentiate those with low physical functioning.

Handgrip maximal strength could not discriminate between older women with and without low physical functioning. This fact may be attributed to the fact that the average handgrip strength in our sample was 21.7 kg, which is higher than the cutoffs applied in sarcopenia and frailty [7,27]. Therefore, it is reasonable to assume that greater grip strength values would not necessarily result in improved physical function without muscle weakness. It is not necessary to have above-average handgrip strength to perform activities of daily living independently.

The results showed no relevance (association or discrimination) of VJ force for physical functioning. One possible explanation for this finding may be related to the nature of the jump task used for assessing muscle force: the participants were asked to perform a single two-legged countermovement jump to maximum height, which outputs a much higher maximum power value but a smaller maximum force [28]. When participants are asked to perform a quick jump (also called escape jump), a higher maximum muscle force is usually recorded [28]. It is possible that with different jumping tasks, different maximum force outputs were observed, and thus the results diverged. However, despite the lack of association found in this study, it is crucial not to undervalue maximal jumping force when analyzing musculoskeletal capacity. However, one study emphasizes lower-limb muscle force more than muscle power to explain the variance in physical functioning assessed by TUG [29]. 

It should be noted, though, that in this study, muscle power was expressed by the height of a countermovement jump (cm) and muscle strength by the time (s) to sit-to-stand five times. The time to conclude the sit to stand task is often used as a fundamental variable within equations to estimate lower-limb muscle power [16]. It will be questionable if this parameter can represent muscle force alone. Thus, the conflicting results may be explained by the type of jump used by our team combined with the different methods of muscle force evaluation. Still, it has been shown that when muscle strength is assessed by lower-body measures, it can influence the sit-to-walk part of the TUG test [30].

Our results suggest that muscle power alone could differentiate older women who reported (questionnaire) or demonstrated (gait speed/agility and daily mobility) low physical functioning from those who did not. Recent publications highlight lower-limb muscle power as a relevant marker for healthy ageing–development/maintenance of functional capacity with a view to well-being in old age more relevant than traditional markers of handgrip strength and muscle mass [23,28].

The importance of muscle power in older adults is well-known, and objectively measured threshold values will help screen those who may be at risk for loss of physical independence. Taking as reference the three markers for physical functionality (perception of physical ability, capacity for mobility, and daily mobility), ~14.0 W/kg, 15 W/kg, and 17.0 W/kg were the cutoff values of muscle power to suspect low physical functionality in which the person’s confidence, capacity, and safety are compromised. Despite the majority of the research on jumping power being carried out in young and athletic populations, Tsubaki et al. (2016) also found that healthy women aged 70–79 had a mean jumping power of 23.0 W/kg [31], while older participants (84.5 ± 4.2 yrs old) in the Osteoporotic Fractures in Men Study had a mean peak power of 20.8 W/kg [32]. These data allow for greater contextualization and understanding of the cutoffs established by our work.

For the assessment of physical functionality, although we resorted to evaluating capacities (CPF, gait/speed/agility) and behaviors (daily mobility), we are aware that the latter is determined not only by capacities but also by motivation and opportunities. In this sense, the muscle power cutoff value indicative of low daily mobility may not be genuinely discriminative of physical functioning. On the other hand, the threshold for low daily steps in community-dwelling older people needs to be defined, which constitutes a limitation of our study. The very low cutoffs defined for TUG and daily mobility in comparison to the adult population normative values may only identify those with very low physical functionality, failing to discriminate against additional older women who are also on the verge of losing physical independence. Additional research must be conducted to understand our cutoff’s usability in different populations and settings. Another limitation of our work is that the usual gait speed was not evaluated, and this parameter was expressed through the TUG. However, we understand that the TUG is highly dependent on the speed and agility of the gait [33], and it is feasible to conclude the participant’s ability to walk. Therefore, acknowledging the relevance of the time spent standing up from the chair and sitting back down, it is possible to conclude the walking ability of the participant.

Despite the limitations mentioned above, the acknowledgment of the relevance of muscle power in old age is rising with the increasing number of scientific publications focusing on muscle capacity [34]. Health and exercise community programs targeting ageing individuals should consider the need to maintain and enhance this capacity. A recent meta-analysis [35] found that high-velocity training, which is specific to enhancing muscle power, greatly affected the physical function of older adults. According to the same study, muscle power-focused training programs appear to have more significant benefits in fast walking speed, timed up-and-go, and the five times sit-to-stand test than traditional resistance exercise. The relationship between maintaining high levels of physical functioning and retaining/regaining of muscle power is strong. This highlights its usefulness in characterizing, understanding, and identifying low levels of physical function and independence in older individuals.

## 5. Conclusions

Musculoskeletal fitness, especially muscle power, can help identify older women with low physical functioning having perceived and demonstrated low physical capacity. More research is needed to understand whether our cutoff values can be used with different populations, such as older men, and how different they are for those who no longer possess the capacity to live independently at home. The decrease in relative jumping power seems to affect first the habitual physical activity, then the mobility capacity, and only later the person’s perception of his physical functioning. That is, the person only realizes he is incapable when his muscle power is already significantly reduced. This highlights the need to assess muscle power in older women prior to the onset of conscious low physical functioning so that programs targeting musculoskeletal capacity can be implemented earlier.

## Figures and Tables

**Figure 1 ijerph-20-05485-f001:**
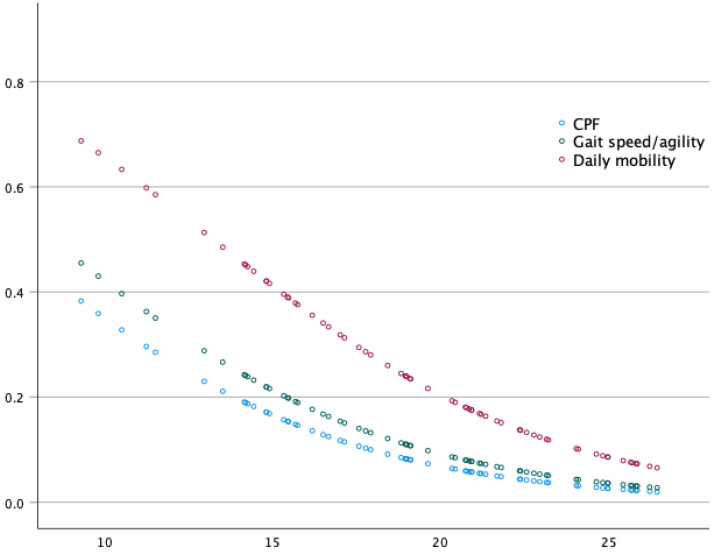
Chance of low physical functioning estimated from a questionnaire (CPF), physical performance (gait speed/agility), or daily mobility (steps/day) according to vertical jump power (W/kg).

**Table 1 ijerph-20-05485-t001:** Characterization of the participants.

	Mean (SD)	Standard Deviation
Age, y	73.62	8.23
Body Mass, kg	66.25	12.33
Body Height, cm	151.59	6.25
BMI, kg/m^2^	19.87	3.05
CPF Score (pts)	19.97	4.49
TUG (s)	6.30	1.86
Daily Steps (n)	6364	2726
Handgrip, kg	21.7	21.7
Rel. Handgrip, Kg/Kg	0.33	0.74
VJ Power, W	1150	340
Rel VJ Power, W/Kg	17.64	4.53
VJ Force, N	1280	290
Rel VJ Force, N/Kg	19.46	3.14
Low CPF (%)	10.6	-
Low TUG (%)	13.6	-
Low Daily Steps (%)	21.2	-

BMI, body mass index; CPF, Composite Physical Function; TUG, Timed Up and Go; Rel, Relative; VJ, Vertical Jump.

**Table 2 ijerph-20-05485-t002:** Associations between musculoskeletal fitness and physical functioning (activities of daily living—ADL, walking speed (TUG time), habitual daily mobility).

Variables	CPF	TUG	Daily Mobility
Handgrip Strength	0.433 **	−0.441 **	0.186
Rel Handgrip strength	0.503 **	−0.292 *	0.252
Jump Power	0.490 **	−0.469 **	0.384 **
Rel Jump Power	0.575 **	−0.371 **	0.460 **
Jump Force	0.062	−0.189	0.390
Rel Jump Force	0.160	−0.01	0.136

Note: * *p* < 0.05; ** *p* < 0.001.

**Table 3 ijerph-20-05485-t003:** Cutoff values for handgrip strength, vertical jump power, and vertical jump force used to identify low physical functioning expressed through CPF (≤14 pts), gait speed/agility (TUG > 8 s), and low daily mobility (PA < 4600 steps/day), classification accuracy, and respective sensitivity and specificity.

	Cutoff	Percentile	AUC (95% CI)	*p*-Value	Se	Sp	Odds Ratio (95% CI) ^a^
	Rel. Handgrip (kg/kg)						
Low Physical Function	-		0.65 (0.43, 0.86)	0.208	-	-	-
Low Gait Speed/Agility	-		0.65 (0.44, 0.87)	0.142	-	-	-
Low Daily Mobility	-		0.64 (0.46, 0.82)	0.115	-	-	-
	Handgrip (kg)						
Low Physical Function	-		0.59 (0.38, 0.80)	0.429	-	-	-
Low Gait Speed/Agility	19.5	32	0.75 (0.57, 0.92)	0.019	55.6	71.9	0.801 (0.670, 0.959)
Low Daily Mobility	-		0.54 (0.36, 0.71)	0.697	-	-	-
	Rel Jump Power (W/kg)						
Low Physical Function	13.9	24	0.79 (0.64, 0.94)	0.013	71.4	81.4	0.790 (0.644, 0.969)
Low Gait Speed/Agility	15.2	35	0.74 (0.56, 0.92)	0.023	66.7	70.2	0.814 (0.682, 0.972)
Low Daily Mobility	17.1	44	0.73 (0.59, 0.87)	0.011	64.3	65.1	0.838 (0.721, 0.974)
	Jump Power (W)						
Low Physical Function	1011	32	0.73 (0.54, 0.93)	0.044	71.4	72.9	0.997 (0.994, 1.00)
Low Gait Speed/Agility	800	18	0.78 (0.59, 0.96)	0.008	66.7	89.5	0.995 (0.992, 0.999)
Low Daily Mobility	-		0.63 (0.47, 0.79)	0.148	-	-	-
	Rel Jump Force (N/kg)						
Low Physical Function	-		0.55 (0.32,0.77)	0.700	-	-	-
Low Gait Speed/Agility	-		0.41 (0.25,0.57)	0.385	-	-	-
Low Daily Mobility	-		0.52 (0.36,0.68)	0.824	-	-	-
	Jump Force (N)						
Low Physical Function	-		0.52 (0.29,0.74)	0.892	-	-	-
Low Gait Speed/Agility	-		0.60 (0.32,0.75)	0.327	-	-	-
Low Daily Mobility	-		0.44 (0.27,0.60)	0.493	-	-	-

^a^ OR (95% CI) from logistic regression.

## Data Availability

The data presented in this study are available on request from the corresponding author. The data are not publicly available due to privacy reasons.

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
