# Peer review of "Musculoskeletal Fitness for Identifying Low Physical Function in Older Women"

_ijerph, 2023, doi:10.3390/ijerph20085485_

Round 1

Reviewer 1 Report

The descriptive statistics are used well!

But you should add more visualizations to your paper and check whether the email name is Gmail or Gmai.

The methods can be improved by introducing much more ways to do a method selection.

Author Response

Dear Reviewer,

Thank you for calling attention to the errors.

Regarding adding more visualizations, we're not sure we understand it well, but we've added a figure at the end of the results section.

However, we do not understand the call to attention regarding the method selection.

Best regards

Reviewer 2 Report

Introduction

- The importance of Physical functioning in the older population was not adequately discussed.

- The need for the study was not adequately stressed.

Methodology

- Sample size calculation needed

- The validity and reliability details of all the outcome measures needed with adequate references.

Author Response

Dear Reviewer,

Thank you for your comments.

We have clarified the meaning and importance of physical function and musculoskeletal fitness.

Both the aim and need for the study have also been reviewed.

In the methodology, information about the sample and the outcome measures was added. It is a convenience sample whose description was added in the methodology.

Validity and reliability references were added.

Best regards

Reviewer 3 Report

Good working! Only two questions, please explain:

What is the relationship between the musculoskeletal fitness and muscle power? Please introduce.

Why the assessment of musculoskeletal fitness are only on the limbs?

Author Response

Dear Reviewer,

Thank you for your comments.

Muscle power, just like muscle strength, is a component of musculoskeletal fitness. It has been clarified in the manuscript.

Regarding the assessments, it is a common practice to evaluate muscle strength and power in the limbs of older people. It happens due to the fact that appendicular muscle mass diminishes greatly with ageing, which in turn is a limitation for the essential daily activities such as walking, squatting, pushing and pulling.

Best regards

Round 2

Reviewer 2 Report

The authors failed to address the following comments

Methodology

- Sample size calculation needed 

- The validity and reliability details of all the outcome measures needed with adequate references. (This data is needed for each outcome measure used in this study)

Author Response

Dear Reviewer,

Our apologies for not being clear enough.

1- No previous sample size calculation was made. We used the data from a large convenience sample with multiple age groups. All women above the age of 65 in the original sample were included in our analysis.

2 - Further corrections to the manuscript have been made to attend to your suggestions.

Best regards.

Reviewer 3 Report

No any questions

Author Response

Thank you for your remarks.